# Cultivation of Spore-Forming Gut Microbes Using a Combination of Bile Acids and Amino Acids

**DOI:** 10.3390/microorganisms9081651

**Published:** 2021-08-02

**Authors:** Sakura Onizuka, Masaru Tanaka, Riko Mishima, Jiro Nakayama

**Affiliations:** Department of Bioscience and Biotechnology, Faculty of Agriculture, Kyushu University, Nishi-ku, Fukuoka 819-0395, Japan; zuzuko4541@gmail.com (S.O.); msr456852@gmail.com (M.T.); r.mishima3519@gmail.com (R.M.)

**Keywords:** gut microbiota, cultivation, spore, germination, co-germinant, bile acid, amino acid

## Abstract

Spores of certain species belonging to Firmicutes are efficiently germinated by nutrient germinators, such as amino acids, in addition to bile acid. We attempted to culture difficult-to-culture or yet-to-be cultured spore-forming intestinal bacteria, using a combination of bile acids and amino acids. The combination increased the number of colonies that formed on agar medium plated with ethanol-treated feces. The operational taxonomic units of these colonized bacteria were classified into two types. One type was colonized only by the bile acid (BA) mixture and the other type was colonized using amino acids, in addition to the BA mixture. The latter contained 13 species, in addition to 14 species of the former type, which mostly corresponds to anaerobic difficult-to-culture *Clostridiales* species, including several new species candidates. The use of a combination of BAs and amino acids effectively increased the culturability of spore-forming intestinal bacteria.

## 1. Introduction

The human intestinal system is inhabited by a wide variety of bacteria. They interact to form complex intestinal microbiota that influence various activities and the health of the host [1,2]. The development of metagenomic sequence technology, which digitizes genetic components of gastrointestinal microbiota, has enabled the discoveries of host-independent metabolic functions of the intestinal microbial community, such as the synthesis of vitamins or neurotransmitters, and the digestion of dietary fibers followed by the fermentative production of organic acids, which play important roles in host energy and immune homeostasis, as well as protection against infection by pathogens [3,4,5,6,7]. However, there are limitations in these culture-independent methods, which are currently mainstream of gut microbiology, for gaining bottom-up insights in the gut microbial community. Further understanding of the roles of each intestinal bacterial member in the community requires the isolation and culture of bacteria from the 4500 species estimated by metagenomic studies, among which 1900 have yet to be cultured [8].

*Clostridiales* is a dominant order in the human large intestine. Species belonging to this order are mostly known as spore-forming species, some of which are substantially involved in human health and disease [9,10,11]. Spores are durable forms of cells, with a completely different structure than vegetative cells, which allows them to survive in unfavorable environmental conditions, such as nutrient deprivation, desiccation, and exposure of strictly anaerobic bacteria to oxygen [12,13,14,15].

The human body harbors two major types of bile acids (BAs), which are cholic acid (CA) and chenodeoxycholic acid (CDCA). Both are synthesized as conjugated forms in the liver, from cholesterol [16]. These BAs are stored in the gallbladder, and released into the duodenum in response to dietary lipids [17,18]. In the digestive tract, residential bacteria deconjugate CA and CDCA. Subsequent dihydroxylation or epimerization at the C-7 position results in secondary BAs that include deoxycholic acid (DCA), lithocholic acid (LCA), and ursodeoxycholic acid (UDCA) [19,20,21]. Fifteen different BA molecules are predominant in the human intestinal tract. Spores of *Clostridium difficile* germinate in response to taurocholic acid (TCA) [22,23,24]. TCA acts as a potent germinator for a variety of spore-forming bacteria, including a number of novel species [9]. Following these studies, we examined the germination-inducing activity of these 15 BA molecules. In addition to TCA, the glycine conjugates GCDCA, GDCA, and GCA displayed potent spore germination-inducing activity. These germination-inducing activities have proven useful for the selective culture of spore-forming bacteria containing difficult-to-isolate species or yet-to-be cultured species. For example, we isolated 72 species containing 10 new species candidates [25].

Spores germinate through a complex process, in response to BAs and to specific environmental factors, such as amino acids (AAs), nucleic acids, and sugars, which are indicators of a suitable environment for growth [26,27]. For instance, efficient germination of *C. difficile* requires a second co-germinant signal, namely, AAs, notably glycine, in addition to BA [14,24,28,29,30]. To sense the signal of those germinants, *C. difficile* has a cascade signaling system, consisting of three Csp proteins (CspA, CspB, and CspC) [31,32,33,34]. These Csp proteases are conserved in many *Clostridia**les*, such as *Clostridiaceae*, *Lachnospiraceae*, and *Peptostreptococcaceae* [35,36]. Further, the germination receptors are diverse, including Ger protein family-sensing AAs in *C. perfringens* and *C. botulinum*, in addition to CspC-sensing TCA in *C. difficile* [37,38]. It is suspected that diverse germination receptors that recognize various environmental signals, such as AAs, as co-germination signals remain to be identified [34,39].

There may be various modes of action for germination of the extensively diverse variety of bacteria that are present in the human intestine, many of which may require both BAs and nutrient germinants, such as AAs, for efficient germination, as has been observed in *C. difficile* [39,40,41]. Therefore, we presume that to enable to culture more diverse spore-forming gut microbes, a combination of BAs and AAs could be used. In this study, we examined the combination of BAs and AAs to expand the spectrum of culturable intestinal bacteria.

## 2. Materials and Methods

### 2.1. Fecal Sample Collection

To investigate the germination activity of the combination of BAs and AAs on bacteria in the gastrointestinal tract, fresh feces were collected from five healthy adults aged 23–35 years, as previously described [25]. The samples were collected in duplicate. One sample was used for bacterial culture in phosphate-buffered saline (PBS). The other sample was collected in RNA later and subsequently used for 16S rRNA amplicon sequencing. Samples collected in PBS were processed within 15 min to maintain the viability of the anaerobic bacteria.

### 2.2. Spore Purification

To obtain spores, fecal samples were processed as described previously [25]. Briefly, the samples were homogenized in PBS followed by the addition of an equal volume of 70% (*v*/*v*) ethanol, and then incubated for 4 h under ambient aerobic conditions at room temperature to kill vegetative cells. The samples were washed three times with PBS and resuspended in PBS for spreading onto the agar medium.

### 2.3. Germination-Inducing Substances

The BAs used in this study included GCA (Tokyo Chemical Industry Co. Ltd., Tokyo, Japan), GCDCA (Nacalai Tesque, Kyoto, Japan), GDCA (Sigma-Aldrich, St. Louis, MO, USA), TCA (Tokyo Chemical Industry Co., Ltd.), and TCDCA (FUJIFILM Wako Pure Chemical Co., Wako, Japan) (structural information of bile acids used in this study is described in Appendix A). GCA, GCDCA, TCA, and TCDCA were dissolved in dimethyl sulfoxide (DMSO) to 5% (*w*/*v*) and filter sterilized. GDCA was dissolved in DMSO to 1% (*w*/*v*) and filter sterilized. To prepare the BA mixture, equal volumes of the five bile acids were mixed.

All AAs were the L-form or glycine. They were purchased from Sigma-Aldrich. Stock solutions of AAs were dissolved in water with pH adjusted to 7.5, with sodium hydroxide (Nacalai Tesque) and sodium chlorite (Sigma-Aldrich). These solutions were sterilized by filtration before use.

### 2.4. Germination and Culturing

The ethanol-treated samples were suspended in PBS to a final volume of 100 µL (containing final concentrations of 30 mM AA and 2% (*v*/*v*) of the BA mixture). To induce germination, they were incubated in an anaerobic chamber containing 10% carbon dioxide, 10% hydrogen, and 80% nitrogen at 37 °C for 30 min and spread on Gifu anaerobic medium (GAM; Nissui Pharmaceutical, Tokyo, Japan) agar plates. The inoculated agar plates were incubated in an anaerobic chamber at 37 °C for 72 h.

### 2.5. 16S rRNA Gene Amplicon Sequencing and Analysis

16S rRNA gene amplicon sequencing and analysis were performed as previously described [25]. Briefly, all colonies grown on the plate were collected and genomic DNA was extracted from the region of the pooled bacterial cell samples using the bead–phenol method [42]. The DNA was used for amplicon sequencing of the 16S V3–V4 by Illumina MiSeq [42]. The microbiota of fecal samples was also determined using bacterial genomic DNA extracted by the bead–phenol method from fecal samples and the V3–V4 amplicon sequencing system in Illumina MiSeq. The obtained sequence data were dereplicated into operational taxonomic units (OTUs) with 97% sequence identity using the Uparse pipeline in Usearch versions 9.2, 10.0. (http://drive5.com/usearch/download.html, accessed on 14 December 2020). The representative sequence of each OTU was subjected to species annotation using EZ biocloud 16S-based ID (https://www.ezbiocloud.net/identify, accessed on 29 June 2021).

### 2.6. Statistical Analyses

Statistical analyses were performed using the R package (V.4.0.5) (https://www.r-project.org/, accessed on 14 December 2020) and Excel 2019 (Microsoft). To compare colony-forming units (CFU) between the treated and control groups, Welch’s t-test or Wilcoxon rank sum test was used for triplicate and quintuplicate experiments, respectively. The operational taxonomic units (OTUs) were clustered based on their relative abundances on the 16 different AA plates. The hierarchical clustering was calculated based on the full concatenation of the Euclidean distance matrix using a heatmap function of the R gplot package.

### 2.7. Accession Number of 16S rRNA Gene Sequences

Raw sequence data were deposited in the DNA Data Bank of Japan (DDBJ) sequence read archive (DRA012257) under BioProject no. PRJDB9975, which contains links and access to stool samples and plate colony data under BioSample SAMD00327995 to SAMD00328050.

### 2.8. Ethics Approval

This study was approved by the ethics committee of the Faculty of Agriculture, Kyushu University (No. 19-006). All methods were performed in accordance with relevant guidelines and regulations. Written informed consent was obtained from donors of fecal samples. We entered and analyzed all samples anonymously and published all data anonymously using personal numbers.

## 3. Results

### 3.1. Effect of Inidividual AAs on the Germination and Colony-Inducing Activities for Spore-Forming Bacteria in Feces

To evaluate the effect of each AA on the germination and colony-inducing activities independently of BAs, the ethanol-treated fecal samples were exposed to various AAs, and then cultured on GAM agar plates. As a positive control, a sample exposed to a mixture of BAs was spread on the agar plate and cultured using the same procedure. An untreated sample was spread on the plate as a negative control. No statistical increase in CFU was observed in the samples that were treated with any AA, while CFU was significantly increased in the positive control treated with the BA mixture compared to the negative control (*p* < 0.01, Welch’s t-test; Figure 1). This result suggests BA-dependent activity of AAs.

### 3.2. Culturing of Spore-Forming Bacteria in Feces Using a Combination of Various AAs with BAs

We next examined the combined effect of AA and BA on the germination/colony induction of spore-forming bacteria in feces. The samples were treated with various AAs, in addition to a mixture of five conjugative BAs (GCA, GCDCA, GDCA, TCA, and TCDCA), which have been indicated to have substantial spore-germinating activity in our previous study [25]. As shown in Figure 2, 12 AAs showed an increasing effect, in which arginine showed statistical significance (*p* < 0.008, Wilcoxon rank sum test). The other 11 AAs increased the CFU, ranging from approximately 1 to 1.5, depending on the samples. However, the differences were not statistically significant.

### 3.3. Classification of Germinating Bacteria by Responsiveness to AAs and BAs

To obtain taxonomic information of the bacteria that germinated and colonized in response to the BA mixture with or without AAs, the colonies grown on each agar plate were pooled and subjected to 16S rRNA gene amplicon sequencing. At the same time, total DNA was directly extracted from the donor’s fecal samples and subjected to amplicon sequencing. Appendix A reports the new OTUs appearing in response to AAs. These new OTUs correspond to approximately 15% of the bacterial population in each donor fecal sample, suggesting the validity of this method to increase the culturability of difficult-to-culture bacteria that are present in the feces. The distribution of OTUs that appeared among the different treatments (Figure 3) was subjected to clustering analysis and it is shown as a heat map. The observed OTUs differed in their reactivity to BA and each AA, and were largely classified into two types. The BA type was found even in the samples that were treated only with BA mixture. The BA–AA type was found only in the samples that were treated with AAs, in addition to the BA mixture. The presence of the BA–AA type suggests the effect of the AAs in expanding the spectrum of culturable species.

### 3.4. Phylogenetic Analysis of Bacterial Species Germinated by the Combination of BAs and/or AAs

Phylogenetic analysis of the BA type and BA–AA type bacterial species was performed. Figure 4 shows the closely related species of the BA type and BA–AA type OTUs. The BA type comprised 14 species, and the BA–AA type comprised 13 species. In both types, the OTUs were mainly distributed to the order *Closridiales*, comprising the family *Lachnospiraceae*, *Clostridiacea*, and *Osillospiraceae*, in addition to other minor Firmicutes groups, such as *Erysipelotrichaceae* and *Bacillaceae*. The BA–AA type contains five novel candidate species, showing < 97% sequence identity in the partial 16S RNA sequence to the closest known species (asterisks in Figure 4), while the BA type contains one novel candidate species.

## 4. Discussion

Isolation and cultivation are required to fully determine the function of each bacterium in the intestine. However, a large proportion of intestinal bacteria, particularly strictly aerobic Firmicutes species, notably classified as order *Clostridiales*, are difficult to culture, or have yet to be cultured. This has hindered the study of intestinal microbiology.

In our previous study, we succeeded in addressing the hurdle of cultural isolation of the *Clostridiales* species, by using the following two aspects of these bacteria: the spore-forming property and the spore-germinating property in response to BAs [25]. In this study, we further enhanced the efficiency of this cultural isolation method using AAs. The protocol of this method is quite simple, and requires only BAs and AAs, in addition to an ordinal anaerobic culture system, but does not require handling of fecal samples under the strictly anaerobic conditions.

In *C. difficile*, glycine is the most effective co-germinating agent. Other molecules, such as L-phenylalanine, L-arginine, L-cysteine, and L-alanine, also induce spore germination when TCA is added [39,40,41]. In contrast, AAs with negative charges on their side chains, such as glutamic acid and aspartic acid, function poorly as co-germinants [41]. In the present study, the number of colonies in the samples in which germination was induced by the germination of a mixture of BAs combined with glutamic acid, was lower than that of other AAs, which is similar to the results obtained in previous studies [41].

In the present study, we also found that there are two types of germination-inducing agents, namely, “BA mixture” and “BA mixture + AA”, based on their different reactivities to germination inducers. Their classification differed at the species level. Earlier studies ranked the germination-inducing activity of various AAs against *C. difficile*, and demonstrated that glycine functioned most effectively as a co-germinant and L-aspartic acid functioned poorly, with a hierarchical ranking of AA co-germination activity [24,39,40,41]. The ranking of this germination-inducing activity was also slightly different among the strains. Other known spore-forming bacteria of the *Bacillus* and *Clostridium* genera have a variety of germination pathways [43,44,45,46]. This difference in reactivity and reaction pathways to germinants is suggested by the different OTUs growing in response to different AAs. Glycine, which effectively induces the germination of *C. difficile* spores, is preferentially used in energy generation by AA metabolism via the Stickland reaction [47]. Arginine, which showed a significant increase in colonies in this study, has been suggested to play an important and regulatory role in the physiology of *Bacillus*, through arginine phosphorylation [48]. Thus, since some AAs are involved in the growth of trophoblast cells, AAs may be involved in the vegetation, as well as germination, of spores. The molecular mechanism on the reception and signal transduction of these germinants and co-germinants, and the following vegetative growth, warrants further studies for a more efficient strategy to address uncultivable spore-forming groups of intestinal bacteria.

The results of the present study suggest that the spectrum of spore-forming bacteria that can be cultured and isolated by BAs can be further expanded by the addition of various AAs. In previous studies, many new candidate species have been successfully isolated by culturing spores with BAs [9,25]. Since there are many unknowns in germination by the combination of BAs and AAs, it would be worthwhile to proceed with large-scale isolation and culture studies using these germinants.

## Figures and Tables

**Figure 1 microorganisms-09-01651-f001:**
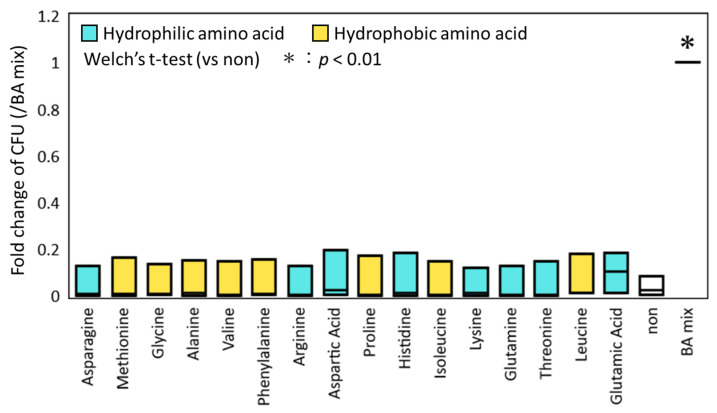
Comparison of fold-change of colony-forming units (CFU) by addition of amino acids (AAs). The ethanol-treated samples collected from three donor subjects were incubated with each AA. The number of colonies that formed on the GAM agar plate was counted after 72 h incubation. Fold-change was calculated based on the bile acid mixture (BA mixture) for each fecal sample and is shown in the box plot. The Welch’s t-test was used to test for significant differences of colony counts between control plate without AA and BA, and plate with AA or BA mixture. The asterisk on the bar represents *p* < 0.01.

**Figure 2 microorganisms-09-01651-f002:**
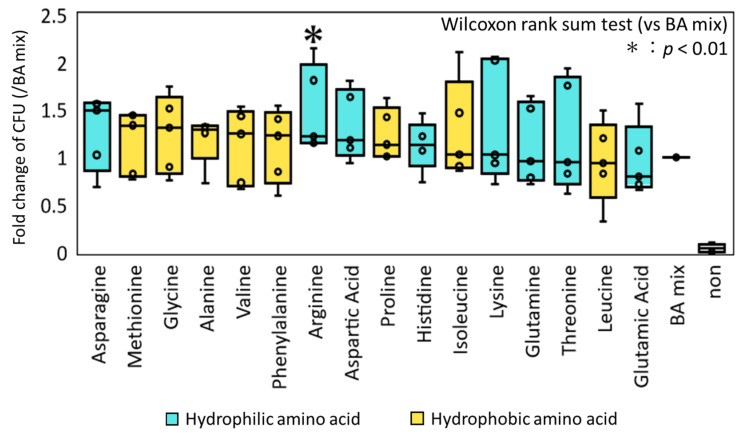
Comparison of fold-change of CFU by addition of AAs to the mixture of BAs. The ethanol-treated samples collected from five donors were incubated with each AA in addition to the BA mixture. The number of colonies formed on the GAM agar plate was determined after 72 h incubation. Fold-change was calculated based on the BA mixture for each fecal sample and is shown in the box plot. The samples were arranged in the order of the median fold-change among the five donors. The Wilcoxon rank sum test was used to test for significant differences between the BA mixture and BA plates. The asterisk on the bar represents *p* < 0.01.

**Figure 3 microorganisms-09-01651-f003:**
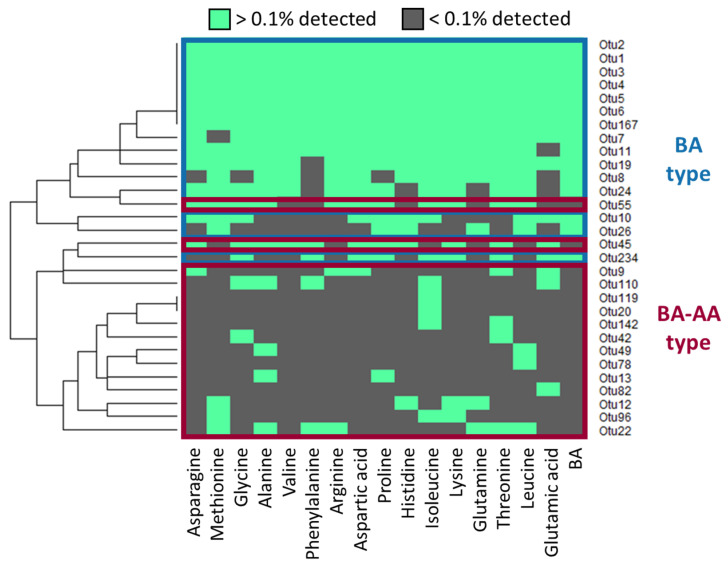
Hierarchical clustering of bacterial operational taxonomic units (OTUs) grown on agar plates supplemented with each AA in addition to the BA mixture. OTUs that accounted for > 0.1% of total population on each plate were selected and subjected to clustering analysis. They are shown in green in the heat map.

**Figure 4 microorganisms-09-01651-f004:**
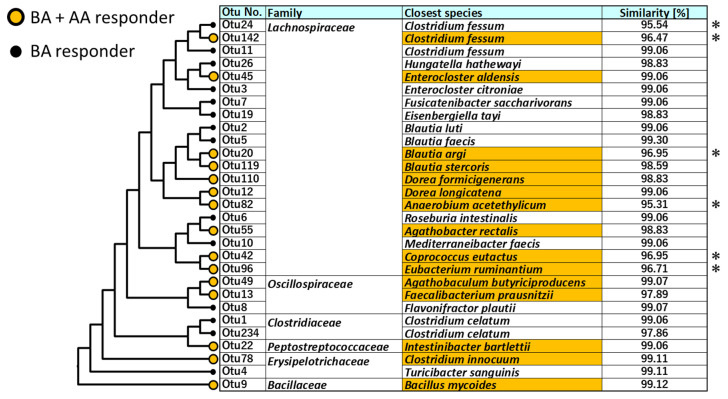
Phylogenetic analysis of BA type and BA–AA type OTUs. The phylogenetic tree was constructed by the neighbor-joining method using the sequences of V3–V4 region of the 16S rRNA gene. Colored OTUs represent BA–AA type. Closest species was identified by homology search of the partial 16S rRNA sequence to EZ biocloud type strain database. Sequence similarities < 97% is marked by asterisk.

## Data Availability

Raw sequence data were deposited in the DNA Data Bank of Japan (DDBJ; https://www.ddbj.nig.ac.jp/index-e.html: accessed on 17 June 2021) sequence read archive (DRA012257) under BioProject No. PRJDB9975, which contains links and access to stool sampling and cultured data under BioSample SAMD00327995 to SAMD00328050.

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
