# Peer review of "Cultivation of Spore-Forming Gut Microbes Using a Combination of Bile Acids and Amino Acids"

_microorganisms, 2021, doi:10.3390/microorganisms9081651_

Round 1

Reviewer 1 Report

In the manuscript, “Cultivation of spore-forming gut microbes using a combination of bile acids and amino acids", Onizuka et al. tested the effect of a combination of bile acids (BAs) and amino acids (AAs) on the cultivation of the spore-forming gut microbes. The authors clearly showed the effect of some AAs depending on the BAs especially on the cultivation of novel species by evaluating CFUs and meta-16S microbiota composition of cultured colonies. The developed procedure will help researchers isolating the yet-to-be cultured strains from the human fecal samples.

In my opinion, overall research strategies are adequate, and additional data indication and discussion will be helpful for readers to conduct the same experimental strategy for isolating objective bacteria from their samples.

Major concern:

  1. Results 3.2 (Figure 2)

Here, only the arginine gave significant effects on the CFU increase. Additional discussion for the effect of arginine is preferable concerning the physiological nature of the increased OTUs by arginine (OTU22 and OTU9), if possible.

  1. Results 3.3 and 3.4 (Figure 3 and 4)

I would like to know the population of the OTUs specifically detected by each AA treatment, especially for the novel species candidates. Comparison of them to those in the fecal microbiota sample used in this study is also informative.

Although such information was shown as higher than 0.1% of the total population of each plate in Figure 3, providing the practical proportions of them in the plate will help to evaluate the possibility of the isolation of new species candidates by using developed procedures.

Minor concern

  1. P1 to P2

Brief structural information on bile acid species that appeared in the text is necessary (e.g., CA, 3α,7α,12α-trihydroxy‑5β-cholanoic acid). I recommend adding that information in the text or preparing a supplementary table for the bile acid species used.

  1. L60

Remove space after "germinants".

  1. L149

I recommend rephrasing that "This result suggests BA-dependent activity of AAs".

  1. Figure legend for Figure 4

Asterisks are not indicated. Please add them in the appropriate places.

[END]

Author Response

Thank you very much for reviewing our manuscript entitled “Cultivation of spore-forming gut microbes using a combination of bile acids and amino acids”. All your comments were helpful to improve this paper. We have revised the manuscript according to your comments. We explain how the manuscript was improved in response to each comment below. We hope the following answers are comprehensively helpful for you to consider publication of this manuscript in Microorganisms.

Major Concern:

  1. Results 3.2 (Figure 2)

Here, only the arginine gave significant effects on the CFU increase. Additional discussion for the effect of arginine is preferable concerning the physiological nature of the increased OTUs by arginine (OTU22 and OTU9), if possible.

[Ans.]

Thank you very much for your suggestion. We added additional discussion for the effect of arginine. (L240-243)

  1. Results 3.3 and 3.4 (Figure 3 and 4)

I would like to know the population of the OTUs specifically detected by each AA treatment, especially for the novel species candidates. Comparison of them to those in the fecal microbiota sample used in this study is also informative.

Although such information was shown as higher than 0.1% of the total population of each plate in Figure 3, providing the practical proportions of them in the plate will help to evaluate the possibility of the isolation of new species candidates by using developed procedures.

[Ans.]

Thank you very much for very critical your suggestion. As you suggested, we draw the new figures showing the population of each OTU as a stacking graph with a control showing the composition OTUs of non-treated donor’s fecal sample. Indeed, the portion of OTUs newly appeared by the addition of AAs. However, However, the OTUs totally corresponds to approximately 15% of population in each donor fecal sample. This suggests the valid it of this method to culture difficult -to-culture bacteria present in the feces at large extent. (L179-182)

Minor Concern

  1. P1 to P2

Brief structural information on bile acid species that appeared in the text is necessary (e.g., CA, 3α,7α,12α-trihydroxy‑5β-cholanoic acid). I recommend adding that information in the text or preparing a supplementary table for the bile acid species used.

[Ans.]

Thank you for your comment. We added supplementary table about structural information on bile acid species accordingly. (L92-93)

  1. L60

Remove space after "germinants".

[Ans.]

We are sorry for the careless mistake. We have corrected that. (L60)

  1. L149

I recommend rephrasing that "This result suggests BA-dependent activity of AAs".

[Ans.]

Thank you very much for helpful comment. We changed the sentences you suggested . (L149)

  1. Figure legend for Figure 4

Asterisks are not indicated. Please add them in the appropriate places.

[Ans.]

We apology this careless mistake. We added asterisks in the appropriate places. (Figure 4)

Reviewer 2 Report

Manuscript by Onizuka et al. reports that a treatment of human faeces with bile salts and amino acids increases the number of spore formers that can be cultivated. The topic is certainly interesting; the possibility to increase the number of cultivable bacteria is essential to understand the biological role of these bacteria in their environment.

The present manuscript is based on a previous work by the same group, in which the Authors evaluated various bile salts and found that some of them increased the number of cultivable bacteria from faecal samples. Here the Authors expand the previous findings by mixing amino acids to the previously tested bile salts.

Although the Author were able to grow 13 species that were not cultivated without aa, the improvements with respect to the previous manuscript are minimal to justify a publication.

I have some questions.

  1. If I understood correctly, the 16S RNA gene sequencing was performed on all colonies grown on plate. It would have been interesting to do the analysis also with the faecal samples. This would have allowed the authors to compare the results of the 16S sequencing with the CFU and assess the % of bacteria that were cultivate +/- aa with respect to the total amount present.
  2. If the aim of the study was to increase the number of cultivable bacteria, why the Author did not use a mix of all amino acids instead of using all aa independently?
  3. The GAM Agar used to grow bacteria after germination-induction is a rich medium, containing peptic digests of animal and plant tissues (therefore, all aa together with many other molecules). Did the Authors tried to use liquid GAM to induce germination?
  4. lines 195-197: the definition of "novel species" only based on a <97% of sequence identity in the fragment of the 16S RNA gene analyzed is not appropriate
  5. I do not understand the point made in the discussion at lines 223-240. As "two types of germination inducers" the Authors mean "BA" and "BA+aa"?

Minor points:

- legend of Figure 4: there are no asterisks in the figure

- lines 229-230: "Other known spore-forming bacteria of the genera Bacillus and Clostridium  ........." should be "Other known spore-forming bacteria of the Bacillus  and Clostridium  genera ........."

- lines 232-233: "This difference in reactivity and reaction pathways to germinants is suggested by the fact in this study that the colonized OTUs differed depending on AAs". please rephrase

- lines 237-240: the sentence here is quite confusing. trophoblast cells? vegetation?

Author Response

Thank you very much for reviewing our manuscript entitled “Cultivation of spore-forming gut microbes using a combination of bile acids and amino acids”. All your comments were helpful to improve this paper. We have revised the manuscript according to your comments. We explain how the manuscript was improved in response to each comment below. We hope the following answers are comprehensively helpful for you to consider publication of this manuscript in Microorganisms.

Q1.

If I understood correctly, the 16S RNA gene sequencing was performed on all colonies grown on plate. It would have been interesting to do the analysis also with the fecal samples. This would have allowed the authors to compare the results of the 16S sequencing with the CFU and assess the % of bacteria that were cultivate +/- aa with respect to the total amount present.

[Ans.]

Thank you very much for the very critical comments. Actually, reviewer 1 also commented same point and we have added supplementary figure showing the comparison of OTU composition of non-treated donor’s fecal sample and OTUs formed from the BA-AA treated fecal samples. Indeed, the population of OTUs newly appeared by the treatment of AA but these OTUs correspond to 15% of bacterial population in the fecal sample. This suggests the validity of this method to culture difficult-to culture bacteria present in the fecal sample at a certain population . (L179-182)

Q2

If the aim of the study was to increase the number of cultivable bacteria, why the Author did not use a mix of all amino acids instead of using all aa independently?

[Ans.]

Thank you for your critical comment. However, there are many amino acids and we thought that some of them have different activity, such as, some are positive and some are negative or so on. So, we thought that it is important to know the activity of each AA at first. Then, as you suggested, we will do the experiment to see the mixed effect of amino acids in next step.

Q3

The GAM Agar used to grow bacteria after germination-induction is a rich medium, containing peptic digests of animal and plant tissues (therefore, all aa together with many other molecules). Did the Authors tried to use liquid GAM to induce germination?

[Ans.]

Thank you very much for reasonable comments. However, we haven’t tested it as the same reason as the comment 2. In this study, we first examined the activity of pure compound. For the next step, we are curious in the mixed effect as you suggest.

Q4

lines 195-197: the definition of "novel species" only based on a <97% of sequence identity in the fragment of the 16S RNA gene analyzed is not appropriate

[Ans.]

You are completely right. This is just suggestive for the candidate of novel species. So, we rephrased the word to “novel candidate species”. (L199-201)

Q5

I do not understand the point made in the discussion at lines 223-240. As "two types of germination inducers" the Authors mean "BA" and "BA+aa"?

[Ans.]

We are sorry to use confusing word. To avoid the confuding, we have rephrasing this part to “there are two types of germination inducing agents, namely “BA mixture” and “BA mixture + AA” (L227-229)

Minor points

  1. legend of Figure 4: there are no asterisks in the figure

[Ans.]

We apology this careless mistake. We added asterisks in the appropriate places. (Figure 4)

  1. lines 229-230: "Other known spore-forming bacteria of the genera Bacillus and Clostridium ........." should be "Other known spore-forming bacteria of the Bacillus and Clostridium genera ........."

[Ans.]

We changed the sentence from “Other known spore-forming bacteria of the genera Bacillus and Clostridium .........” to “Other known spore-forming bacteria of the Bacillus and Clostridium genera .........”. (L234)

  1. lines 232-233: "This difference in reactivity and reaction pathways to germinants is suggested by the fact in this study that the colonized OTUs differed depending on AAs". please rephrase

[Ans.]

We changed the sentence from “This is because they recognize structurally different germinants by encoding multiple germination receptors” to “This difference in reactivity and reaction pathways to germinants is suggested by the fact in this study that the colonized OTUs differed depending on AAs”. (L235-237)

  1. lines 237-240: the sentence here is quite confusing. trophoblast cells? vegetation?

[Ans.]

We are sorry for the careless use of incorrect word. We corrected the term from “Vegetation process” to “vegetative growth”. (L246)

Reviewer 3 Report

The study by Onizuka and co-authors proposes new cultivation procedure of the spore-forming Firmicutes phylotypes by the combination of AAs and BAs (but not by BAs alone). The reviewer think that the study is interesting and that the use of both AAs and BAs is novel. While the manuscript can be further improved, as suggested below.

Major comment
1.    The aim of this study is not well explained in Introduction section. The author should explain that there are number of spore-forming gut microbes that are not germinated by using bile acids only. C. difficile spore is known to be germinated by the combination of BAs and glycine. Therefore, the authors (may) hypothesized that they can culture more divers spore-forming gut microbes by the combination of BAs and AAs.

2.    In this paper, authors used a mixture of 16 amino acids and bile acids to isolate spore-forming bacteria. Why did you not use the other four amino acids (serine, tryptophan, etc.)?

3.    In Fig. 2, many colonies were formed on GAM-arginine agar. On the other hand, less OTUs were observed with GAM-arginine agar in the experiment shown inFig. 3. How do you explain?

4.    In my point of view, OTU55 and OTU45 belongs to “BA-AA type”. does the author define otu55 and otu45 as BA type even though their why detection rate is less than 0.1% when germination is induced by bile acid alone?

Specific comments #1 (Figure 2)
  “BA mix” should be located in 2nd right position.

Specific comments #2 (Figure 4)
  I cannot find asterisk in Fig. 4, even though there are description “Sequence similarities <97% is marked by asterisk” in footnote. Add the asterisk ore delete the explanation.

Specific comments #3
  I understand the important role of glycine on C. difficile spores germination; however, there are seven descriptions on the effects of glycine on C. difficile in this manuscript. I think it’s too much for the readers. Please consider.

Remarks (the author don’t have to do further experiments nor revise the manuscript)
  In this study, the authors add single amino acids for germination. Why the author didn’t use a mixture of amino acids? How do you think that there might be the combination of two or more amino acids are essential for some OTUs?

Author Response

Thank you very much for reviewing our manuscript entitled “Cultivation of spore-forming gut microbes using a combination of bile acids and amino acids”. All your comments were helpful to improve this paper. We have revised the manuscript according to your comments. We explain how the manuscript was improved in response to each comment below. We hope the following answers are comprehensively helpful for you to consider publication of this manuscript in Microorganisms.

Major Comment

  1. The aim of this study is not well explained in Introduction section. The author should explain that there are number of spore-forming gut microbes that are not germinated by using bile acids only. C. difficile spore is known to be germinated by the combination of BAs and glycine. Therefore, the authors (may) hypothesized that they can culture more divers spore-forming gut microbes by the combination of BAs and AAs.

[Ans.]

Thank you very much for the comment. We will add one sentence to clearly explain the aim of this study in L71 to L73.

  1. In this paper, authors used a mixture of 16 amino acids and bile acids to isolate spore-forming bacteria. Why did you not use the other four amino acids (serine, tryptophan, etc.)?

[Ans.]

Regarding tryptophan and tyrosine, we omitted them due to insolubility to aqueous buffer at neutral pH that does not affect the result for CFU. Regarding serine, we examined the effect for the CFU and found it more or less induce colony formation from spore similarly to the other amino acids. However, unfortunately, we failed to do MiSeq sequencing and could not obtained OTU data. That is why we did not show the result in Fig. 2 and 3.

  1. In Fig. 2, many colonies were formed on GAM-arginine agar. On the other hand, less OTUs were observed with GAM-arginine agar in the experiment shown in Fig. 3. How do you explain?

[Ans.]

Actually, Figure 2 shows the number of colonies while Figure 3 shows the variety of OTUs. Therefore, these data show that arginine induced certain taxonomic groups of OTUs represented in Fig 3 heatmap efficiently.

  1. In my point of view, OTU55 and OTU45 belongs to “BA-AA type”. does the author define otu55 and otu45 as BA type even though their why detection rate is less than 0.1% when germination is induced by bile acid alone?

[Ans.]

Thank you very much for the right comment. We have corrected these two OTUs as BA-AA type in Figure 3 and Figure 4.

Specific comments #1. (Figure 2)

  “BA mix” should be located in 2nd right position.

[Ans.]

Thank you very much for the comment. We rewrote Figure 2 accordingly.

Specific comments #2 (Figure 4)

I cannot find asterisk in Fig. 4, even though there are description “Sequence similarities <97% is marked by asterisk” in footnote. Add the asterisk ore delete the explanation.

[Ans.]

We apology this careless mistake. We added asterisks in the appropriate places. (Figure 4)

Specific comments #3

I understand the important role of glycine on C. difficile spores germination; however, there are seven descriptions on the effects of glycine on C. difficile in this manuscript. I think it’s too much for the readers. Please consider.

[Ans.]

Thank you very much for the comment for the improvement of this manuscript. According to this comment, we deleted the description that C. difficile germinates in response to glycine in the last paragraph of introduction.

Remarks (the author don’t have to do further experiments nor revise the manuscript) In this study, the authors add single amino acids for germination. Why the author didn’t use a mixture of amino acids? How do you think that there might be the combination of two or more amino acids are essential for some OTUs?

[Ans.]

Thank you very much for your striking comment. In this manuscript, we felt it priority to know the activity of each amino acid, because there are many amino acids, each of them may have different activity. Once we know the effect of each amino acid, we may design the cocktail of best mix. So, as a next step, we are curious to examine their mixed effect as you suggest.

Round 2

Reviewer 2 Report

Please change:

"At the same time, 16S rRNA was directly extracted from donor’s fecal samples and subjected to the amplicon sequencing. As shown in Supplementary Figure S1, the portion of OTUs newly appeared by the addition of AAs. However, these OTUs totally corresponds to approximately 15% of population in each donor fecal samples, suggesting the validity of this method to culture difficult-to-cultured bacteria present in the feces at a certain extent."

into:

"At the same time, total DNA was directly extracted from donor’s fecal samples and subjected to the amplicon sequencing. Supplementary Figure S1 reports the new OTUs appearing in response to AAs. These new OTUs corresponds to approximately 15% of the bacterial population in each donor fecal samples, suggesting the validity of this method to increase the culturability of difficult-to-culture bacteria present in the feces."

Please change:

"This difference in reactivity and reaction pathways to germinants is suggested by the fact in this study that the colonized OTUs differed depending on AAs."

into:

"This difference in reactivity and reaction pathways to germinants is suggested by the different OTUs growing in response to different AAs."

Author Response

Thank you very much again for reviewing our manuscript entitled “Cultivation of spore-forming gut microbes using a combination of bile acids and amino acids”. According to the comment, we changed the indicated sentence in the results part (L184-189) and the conclusion part (L243-245). We acknowledge this suggestion that definitely improved our paper.  We hope that the manuscript is now acceptable for the special issue in Microorganisms.